# Preliminary Studies on Genetic Profiling of Coffee and Caffeine Consumption

**Roseane M. Santos** 

Department of Pharmaceutical Sciences, South University School of Pharmacy, Savannah, GA 31406, USA; rsantos@southuniversity.edu; Tel.: +1-912-201-8131

**Abstract:** Regular coffee intake has been associated with reduced risk of developing serious chronic diseases. The hypothesis of this study is that coffee consumers present a particular pattern/trend of genotypes that ultimately will shed light on new gene targets to treat the diseases, from which regular coffee intake has preventive effects. Sixteen SNPs identified at genome-wide association studies (GWAS) on coffee and caffeine consumption were genotyped using real-time restriction-fragment length polymorphism-polymerase chain reaction (RT-PCR). The DNA samples were the same from a previous pilot study where 15 healthy volunteers donated two blood samples collected before and after drinking a standard cup of coffee and had caffeine plasma levels and CYP 1A2 genotype (rs762551) determined. The cross-examination of the data showed that six of the sixteen SNPs exhibited a negative allelic effect direction and nine of them showed a positive effect direction of which three of them had results confirmed by a recent GWAS. There is a need of a more in-depth study to understand the effects of the presence or absence of specific variant alleles as players to benefit the health of coffee consumers.

**Keywords:** coffee; caffeine; genotype; consumption; traits; association

## 1. Introduction

Coffee is the most widely consumed beverage in the world with known health benefits [1]. Besides caffeine (0.5–1.0% of green coffee beans) [2], a well-known CNS stimulant, coffee contains a very complex mixture of organic compounds, such as chlorogenic acids, caffeic acid, kahweol, trigonelline, and minerals. However, roasting coffee gives rise to more healthy compounds, known as quinolactones, as well as some other polycyclic aromatic hydrocarbons with carcinogenic effects [3]. Caffeine is primarily metabolized by human hepatic microsomal reaction of 3-demethylation, catalyzed by CYP1A2, which is responsible for approximately 95% of caffeine metabolism [4]. CYP1A2 DNA code is located at 15 q24 and exhibits polymorphism that can determine a decrease in the enzyme's inducibility. Individuals presenting with homozygous variant CYP1A2 (rs762551-C allele) are slow caffeine metabolizers, whereas individuals who are carriers of CYP1A2 (rs762551-A allele) are fast metabolizers [5,6]. In a previous pilot study, we aimed to determine if the genetic variability of caffeine metabolism could influence coffee consumption. The study showed that 8 out of 11 healthy volunteers (two samples were mishandled and were not tested) presented a fast metabolizer phenotype and displayed a large variability in their caffeine levels (0–0.67 mg/L). One volunteer presented a slow metabolizer phenotype and the highest caffeine blood levels (1.1 mg/mL) [7]. The study confirmed the relationship between genotype/phenotype and blood levels of caffeine, as well as the prevalence of the fast metabolizer phenotype in the population (see column 1 for caffeine plasma levels and SNP # 9 for caffeine metabolism genotype on Figure 1). However, the study could not reach any conclusion about the relationship between genotype and coffee consumption due to the small sample size.

Linked disequilibrium (LD) was first used in 1960 signifying the presence of non-random association of alleles at two or more loci in the same chromosome [8]. There are 3 possible genotypes: homozygote, homozygote-variant, and heterozygote, according with the predefined major and minor allele. The ancestral allele is normally considered the major allele and the variant allele is considered the minor allele. The discovery of haplotype blocks, non-overlapping sets of loci in strong LD, led to a worldwide effort to identify SNPs in haplotype blocks in the human genome—the International HapMap Project. The first trial identified over a million SNPs and the second generation characterized 3.1 millions of SNPs in the same original group of individuals [9]. Genomic Wide Association Studies (GWAS) test sets of SNPs associated with a specific condition comparing allele frequencies on affected and non-affected individuals [10]. Coffee Consumption is a very complex trait as it involves many sets of SNPs' loci, in different chromosomes displaying increased or decreased inducibility in the phenotypic expression of various traits.

Recently, The Coffee and Caffeine Genetics Consortium was created with the purpose of using genome-wide association studies (GWAS) to identify specific loci in the genome associated with coffee and caffeine consumption. Their first results point to eight loci that show genome-wide significance. Six of these are located in or near genes potentially involved in the pharmacokinetics (ABCG2, AHR, POR, and CYP1A2) and pharmacodynamics of caffeine (BDNF and SLC6A4) [11]. Therefore, genetic factors could be a valuable tool to study the potential health effects of coffee by means of gene–environment interaction [12]. Another GWAS on coffee and caffeine consumption [7,11,13,14] identified SNPs at the aryl hydrocarbon receptor region (AHR) and between the CYP1A1 and CYP1A2 gene regions that present significant association with habitual caffeine and coffee consumption. According to Josse et al. [14,15], the 23-kbp segment between CYP1A1/CYP1A2 displays a SNP (rs2472297-T-allele) associated with increased caffeine intake. They also found that this intergenic locus at 7 p21, which corresponds to the aryl hydrocarbon receptor (AHR), has a regulatory role in basal and substrate-induced expression of CYP1A1 and CYP1A2. They concluded that it is possible that genotypes associated with increased CYP1A2 enzyme activity resulted in increased caffeine metabolism and possibly increased caffeine/coffee consumption.

In addition, other studies [14,16] found significant evidence of association with coffee and caffeine consumption at NRCAM gene, which is implicated with addictive behavior and other independent hits such as CPLX3-ULK3 (caffeine and blood pressure), CSK-NCALD (addictive behavior), and CHRNA3 (lung cancer and smoking traits) regions.

The objective of the present study is to examine the relationship between 16 SNPs found to be significantly associated with coffee and caffeine consumption through GWAS within a small sample of volunteers. Our hypothesis is that regular coffee consumers might display similar genotype patterns (positive or negative allele effect) for those sixteen SNPs and this could lead us to potential new targets to treat/prevent the various chronic disorders for which regular coffee consumption has preventive effects.

## 2. Materials and Methods

### 2.1. Sample Population

The graphical abstract displays the workflow followed in this study. The sample population is composed of 13 healthy volunteers that donated DNA samples from a previous pilot study [7]. The pilot study consisted of random enrollment of healthy volunteers within the graduate student population who agreed to participate and provided written informed consent. On the morning of the study, the volunteers presented to the clinic with a minimum of 8 h fasting and having drawn a baseline blood sample (zero-point blood sample). A standard breakfast that included a cup of coffee (150 mg of caffeine) was administered and timed. The plasma levels of caffeine peak on average between approximately 45–60 min after drinking a cup of coffee, [1]; at which time, another blood sample was collected (post-coffee sample). The volunteers also completed a brief food and beverage questionnaire as part of the research protocol. The caffeine plasma levels before and after drinking a cup of coffee

are determined following the HPLC method of Blanchard J and col. [17]. Briefly, the mobile phase is a solution composed of a 10 mM acetic acid/acetate buffer, pH 4.0, 85/15, *v/v*, buffer/acetonitrile. A standard curve ranging from 10 to 0.078 µg/mL of caffeine was used for external quantification. Two hundred and fifty microliters of plasma obtained from each of the blood samples is vortexed for 10″ and centrifuged at 12,800 *g* for 10 min. The supernatant is filtered and 10 µL of the filtrate is injected into a C18 reversed-phase column attached to HPLC system (Shimadzu Model LC-20AT, Prominence, Kyoto, Japan). The HPLC method is isocratic (flow of 1.0 mL/min), using an UV detector at 273 nm wavelength scanned over 10 min.

### 2.2. DNA Preparation and Genotyping

The DNA samples are prepared from 2.0 mL of whole blood collected in EDTA tubes following the procedure described on the previous pilot study [7]. Sixteen SNPs within the highest hits from GWAS on coffee and caffeine consumption are the SNPs selected for this study (Table 1). The SNPs were detected by real-time restriction-fragment length polymorphism-polymerase chain reaction (TaqMan Genotyping Assay, Applied Biosystems, Carlsbad, CA, USA). The assays run in duplicate using StepOne Plus Real-time PCR (Applied Biosystems, Carlsbad, CA, USA) and the data analyzed using TaqMan Genotyping Software.

**Table 1.** Localization, gene names, and variant allele of all 16 SNPs studied.

| SNP # | Marker (rs) | EA/NEA | CHR | Gene Symbol | Position Kb | Gene Name |
|---|---|---|---|---|---|---|
| 1 | 2470893 | [C/T] | 15 q24 | CYP1A1-1A2 region | 74727108 | *Cytochrome P450 family 1A1* |
| 2 | 2472297 | [C/T] | 15 q24 | CYP1A1-1A2 region | 74735539 | *Cytochrome P450 family 1A1* |
| 3 | 6495122 | [A/C] | 15 q24 | CPLX3/ULK3/ | 74833304 | *Unc-51 like kinase 3 Complexin 3* |
| 4 | 2472304 | [G/A] | 15 q24 | CYP 1A2 | 74751897 | *Cytochrome P450 family 1A2* |
| 5 | 1378942 | [A/C] | 15 q24 | CSK | 74785026 | *c-src tyrosine kinase microRNA 4513* |
| 6 | 12148488 | [G/T] | 15 q24 | PPCDC/SCAMP5 | 75090201 | *Secretory carrier membrane protein 5; phosphor-pantothenoylcysteine decarboxylase* |
| 7 | 4410790 | [C/T] | 7 p21 | AHR region | 17244953 | *Aryl Hydrocarbon Receptor* |
| 8 | 6968865 | [A/T] | 7 p21 | AHR region | 17247645 | *Aryl Hydrocarbon Receptor* |
| 9 | 762551 | [C/A] | 15 q24 | CYP 1A2 | 74749576 | *Cytochrome P450 family 1A2* |
| 10 | 3761422 | [C/T] | 22 q11 | SPECC 1L/ADORA2 | 24430704 | *Adenosine Receptor 2A* |
| 11 | 9526558 | [A/G] | 13 q14 | CAB39L | 49408376 | *Calcium Binding protein g39-like* |
| 12 | 1051730 | [A/G] | 15 q25 | CHRNA3 | 78601997 | *Cholinergic receptor nicotinic alpha 3/alpha 5 (neuronal)* |
| 13 | 2066853 | [A/G] | 7 p21 | AHR | 17339486 | *Aryl Hydrocarbon Receptor* |
| 14 | 382140 | [A/G] | 7 q31 | LAMB4/NRCAM | 108141755 | *Neuronal cell adhesion molecule* |
| 15 | 16868941 | [A/G] | 8 q22 | NCALD | 102040149 | *Neurocalcin Delta* |
| 16 | 17498920 | [A/G] | 8 q22 | NCALD | 102043861 | *Neurocalcin Delta* |

EA, Effect allele; NEA, Non-effect allele; CHR, Chromosome.

## 3. Results

Our findings are summarized in two tables and one figure. Figure 1 is a novel format to display results from genotyping data of SNPs. The figure shows the results of each SNP genotyped displayed in a color-coded trait. This display is named as "genotypogram"; to the best of our knowledge, this is the first time that genotypes and associated traits have been displayed under this unique format. In this new format, the SNPs are arranged on the genotypogram according to their direct or positive association (similar pattern) or inverse or negative association (opposite pattern) between the results from the 13 volunteers (homozygous variant, homozygous major allele, or heterozygous). Each trait had a specific color selected and black color used for unknown traits (the label of Figure 1 is the last 2 rows of the table). The bright colors related to homozygous variant genotype, light colors related to homozygous major allele, and white for heterozygous genotype. A variant allele is the minor allele, in which the single nucleotide substitution is present. The wild-type allele represents the major allele. Information about the position of the SNP in the chromosome, number of mutants per SNP, and

number of mutants per volunteer is in Figure 1. The last two rows describe the color code for each trait and genotype.

It is interesting to notice the influence of one SNP located at one chromosome upon another SNP located at another chromosome. For example, SNP 13 at chromosome 7 p21 induces the transcription of CYP1A2 at chromosome 15 q24 due to a stimulus of the dioxin response element (DRE) at the promoter region of that gene. This is possible because we now know that chromosomes are not at a stationary position inside the nucleus, but rather reside in different parts of the nucleus known as chromosome territories (CTs) depending on their activity. In addition, the architecture of the nucleus appears to change with aging, disease status, or as our needs shift (16–18).

Observing Figure 1, thirteen of the 16 SNPs studied (13/16) had a variant genotype in the sample population (*n* = 13) (refer to the last rows of the figure under # homozygote variant per SNP), ranging from nine volunteers to one. Therefore, we made a scale of frequency for the presence of the variant allele in this group ranging from very high (8–9), high (6–7), moderate (4–5), low (2–3), to very low (0–1) number of volunteers with variant genotype in the study sample.

Table 2 is a summary of the genotypes' frequency found in the study population: homozygote variant, homozygote (wild type), and heterozygote. It displays, in percentage and number of volunteers, the frequency of each possible genotype for all the sixteen SNPs in the study. The observation of those percentages allowed us to suggest a potential direction of the variant allele effect in relation to the wild-type allele. The allelic effect direction is the direction (add or subtract) that the contribution of the effect allele (variant) has on a phenotype, increasing (positive) or decreasing (negative) the presence of the wild type allele phenotype in the population [18].

In the intergenic region between CYP1A1 and CYP11A2 displays SNPs 1 and 2 (rs2470893 and rs2472297). The study found a trend towards predominance of the variant allele, considering that both SNPs had only 1 volunteer homozygous for the wild-type allele, suggesting that the wild-type allele might be the recessive one (Table 2).

SNP 3 (rs6495122) is localized in between two genes, CPLX3 and ULK3, running in opposite directions, whereas SNP 4 (rs2472304) is located at CYP1A2 gene; both are in the chromosome 15 q24. The data on Figure 1 shows a strong positive association between SPNs 3 and 4, which the number of volunteers are quite the same for each of the three possible genotypes, suggesting a potential direct association between those SNPs with predominance of the wild-type allele.

SNP 3 and 5 (rs6495122 and rs1378942) are located in the same arm of chromosome 15 in close proximity (see localization on Table 1). The data on Table 2 shows an inverse relationship between the genotypes, in which SNP 3 has a trend towards wild-type allele predominance and SNP 5 displays similar trend towards the variant allele.

SNPs 4 and 5 present a strong inverse relationship between genotypes, which SNP 4 predominates the wild-type allele and SNP 5 the variant allele within the exact the same group of volunteers (see Figure 1 and Table 2).

SNP 5 is localized at the CSK gene while SNP 6 is located in between PPCDC and SCAMP5 genes, both running forward in the chromosome 15 q24 position. The data on Table 2 show a trend toward predominance of the variant allele when comparing SNPs 5 and 6. Both SNPs presented a high to very high frequency of variant allele, 8/13 and 6/13, for SNPs 5 and 6, respectively.

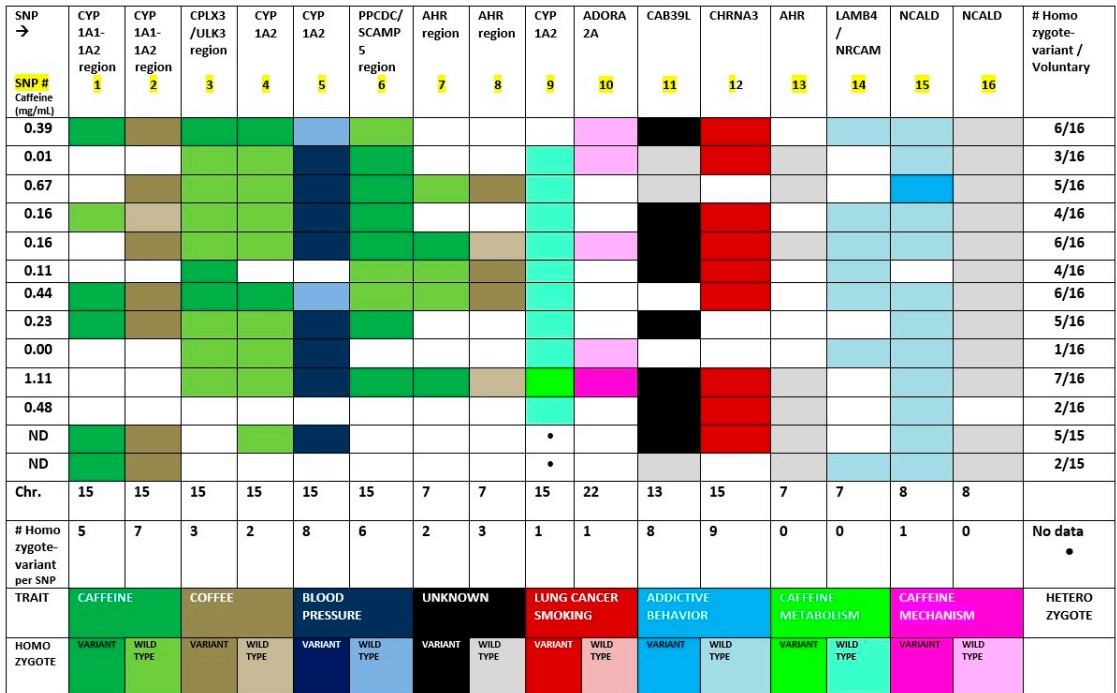

**Figure 1.** Genotypogram of coffee and caffeine consumption and related traits: 16 SNPs localized at five chromosomes, genotyped from 13 healthy volunteers.

SNPs 7 and 8 are located in the promoter region of the AHR gene (where SNP 13 is located). All three variant alleles (SNPs 7, 8, and 13) have a very low to low frequency in the group: two, three, and zero, respectively. SNPs 7 and 8 share the same group of volunteers presenting a heterozygous genotype but the data is not conclusive in respect to which allele is the predominant in these two SNPs.

SNPs 9 and 10 represent caffeine metabolism (CYP1A2) and caffeine mechanism of action (ADORA2A), respectively. The two SNPs showed a very low variant allele frequency in the group (1/13) and were present in the same volunteer. All the others were either wild-type or heterozygous.

SNPs 11 and 12 presented a strong potential positive association with predominance of the variant allele. We have not identified a wild-type genotype for SNP 12 in this group as they were all either homozygote-variant or heterozygous. It was the SNP with the highest frequency of variant allele in the population studied with 9 of the 13 volunteers showing a homozygote-variant genotype.

SNPs 13–16 were all either wild-type or heterozygous, except for SNP 15, which presented one homozygote-variant in the whole group. The frequency of variant alleles for these four SNPs was 0, 0, 1, and 0, respectively.

SNPs 13 and 14 had the same number of heterozygous genotype (6/13) and same number of wild-type (7/13) and there were no homozygote-variant genotype in this study group. We assume that wild-type is the most common genotype for those SNPs.

SNPs 15 and 16 presented almost all wild-type genotype (11/13 and 12/13, respectively), showing a tendency for wild-type genotype for those SNPs as well. Overall, all four SNPs (13–16) displayed similar pattern in their genotype data.

**Table 2.** Potential Variant Allele direction effect for each of the 16 SNPs analyzed.

| SNP # | Homozygote Variant % Frequency | Homozygote Wild-Type % Frequency | Heterozygote % Frequency | Variant-Allele Effect Direction | Marker | Trait (NCBI) |
|---|---|---|---|---|---|---|
| 1 | 38.5 (5/13) | 7.7 (1/13) | 53.8 (7/13) | Subtract (Negative) | CYP1A1-1A2 region | Caffeine/Addictive behavior |
| 2 | 53.8 (7/13) | 7.7 (1/13) | 38.5 (5/13) | Subtract (Negative) | CYP1A1-1A2 region | Coffee |
| 3 | 23.1 (3/13) | 53.8 (7/13) | 23.1 (3/13) | No effect | CPLX3/ULK3/ | Caffeine, blood pressure; addictive behavior |
| 4 | 15.4 (2/13) | 61.5 (8/13) | 23.1 (3/13) | Add (Positive) | CYP 1A2 | Caffeine |
| 5 | 61.5 (8/13) | 15.4 (2/13) | 23.1 (3/13) | Subtract (Negative) | CSK | Blood Pressure |
| 6 | 46.2 (6/13) | 23.1 (3/13) | 30.8 (4/13) | Subtract (Negative) | PPCDC/SCAMP5 | Caffeine |
| 7 | 15.4 (2/13) | 23.1 (3/13) | 61.5 (8/13) | Add (Positive) | AHR region | Caffeine |
| 8 | 23.1 (3/13) | 15.4 (2/13) | 61.5 (8/13) | Add (Positive) | AHR region | Coffee |
| 9 * | 9.0 (1/11) | 81.8 (9/11) | 9.0 (1/11) | Add (Positive) | CYP 1A2 | Caffeine PK |
| 10 | 7.7 (1/13) | 30.8 (4/13) | 61.5 (8/13) | Add (Positive) | SPECC 1L/ADORA2A | Caffeine PD |
| 11 | 61.5 (8/13) | 30.8 (4/13) | 7.7 (1/13) | Subtract (Negative) | CAB39L | Unknown |
| 12 | 69.2 (9/13) | 0.0 | 30.8 (4/13) | Subtract (Negative) | CHRNA3 | Lung cancer Smoking |
| 13 | 0.0 | 53.8 (7/13) | 46.2 (6/13) | Add (Positive) | AHR | Unknown |
| 14 | 0.0 | 53.8 (7/13) | 46.2 (6/13) | Add (Positive) | LAMB4/NRCAM | Addictive behavior |
| 15 | 7.7 (1/13) | 84.6 (11/13) | 7.7 (1/13) | Add (Positive) | NCALD | Addictive behavior |
| 16 | 0.0 | 92.3 (12/13) | 7.7 (1/13) | Add (Positive) | NCALD | Unknown |

\* *n* = 11.

## 4. Discussion

Figure 1 shows that five SNPs exhibited high to very high frequency (6–9 out of 13 subjects) in the group. Only one SNP presented a moderate frequency rank (4–5 out of 13 subjects) and the remainder 10 SNPs had a low to very low frequency (0–3 out of 13 subjects) of homozygote variant-allele in the group. The SNPs that had high to very high frequency rank suggest a negative effect direction (SNPs 1 and 2, 5 and 6, 11 and 12). On the contrary, SNPs with low to very low rank of frequency suggest positive effect direction (SNPs 3, 4, 7, 8, 9, 10, 13, 14, 15, and 16).

No correlation was detected between the genotype and levels of caffeine in the blood after an acute dose of caffeinated coffee. However, we observed that the two volunteers with the lowest levels of caffeine (0 and 0.01 mg/mL), who presented a fast metabolizer phenotype (CYP1A2-rs762551-A allele) for caffeine in the pilot study, also showed a low to very low frequency of homozygote variant allele for all the 16 SNPs associated with coffee and caffeine consumption. In contrast, the only volunteer that had the highest level of caffeine in the previous pilot study (1.1 mg/mL) and the only one with a slow metabolizer phenotype for caffeine (CYP1A2-rs762551-C allele), showed the highest number of homozygote-variant SNPs in the group studied (7/16). It is tempting to assume that a slow caffeine metabolizer phenotype is associated with an increased number of homozygote-variant for all the sixteen SNPs associated with coffee consumption genotyped.

It is an interesting finding, observing the bright colors that represent the homozygote-variant of the SNPs on Figure 1, the presence of matching patterns (for ex. SNPs 1 and 2, 3 and 4, 5 and 6, 11 and 12, 15 and 16) and also the opposing pattern such as 7 and 8.

Our challenge now is to dig into this new information to better understand the intricate effect of the variant alleles and theirs associated traits as they relate to beneficial or harmful health consequences of drinking coffee regularly.

It is important to highlight that many of these SNPs were recently identified through GWAS and, consequently, not much is known about them. The statistical analysis itself does not provide information about why alleles at different loci are non-randomly associated [8].

## 5. Conclusions

The crosstalk between distant genes in the genome is now starting to be better understood due to new cellular techniques to look into nuclear architecture. The emerging view is that chromosomes are compartmentalized into discrete chromosome territories (CTs), which are non-randomly assigned. The location of the gene within a CT seems to influence its access to the machinery responsible for specific nuclear functions, such as transcription and splicing [19–21]. Chromosome conformation capture (3C) techniques offer detailed views of the associations among distant genomic loci (16), thus confirming our intra- and inter-chromosomal SNPs relationships.

The strength of this study is the possibility to cross-examine the information about genotypes from the same subjects on 16 SNPs previously identified as significantly expressed during association studies of coffee and caffeine consumption. The weakness of our study is the size of the population studied. However, to some extent, the fact that the sample population was randomly selected between college students empowers our data, which is confirmed by recent GWAS of coffee consumption in the Japanese population (J-MICC Study) [22]. The study found that SNPs 7 and 8 (AGR3-AHR intergenic region) showed a positive effect direction (Table 2). The other common SNP studied was LAMB4-NRCAM (SNP 14) that also confirmed our data with a positive effect direction. Another SNP also studied in our pilot is at CPLX3-ULK3 intergenic region (SNP 3) is shown to have a negative effect direction, which we could not determine an effect in our sample.

This study with a small group of volunteers could capture the allelic effect direction of the majority of the sixteen SNPs analyzed. Six of them displayed a negative effect and nine of them displayed a positive effect. Additionally, the interesting finding that the fast caffeine metabolizer subject presented the smallest number of homozygote-variant alleles as oppose to the slower caffeine metabolizer that showed the higher number of homozygotes-variant alleles.

Our next step is to reproduce this experiment with a broader sample of coffee consumers and compare the results with non-coffee consumers. In addition, we will add a few more SNPs that were identified after this project started, such as the ones localized at ABCG2, POR, and BDNF, and LC6A4 genes related to the pharmacokinetics and pharmacodynamics of caffeine, respectively.

## 6. Patents

A provisional patent was filled to secure the new format to display genotypes and traits using a color-coded system named genotypogram.

**Funding:** This research received no external funding.

**Acknowledgments:** The authors would like to acknowledge Valerie Yaughn, Director of the Library at South University, Savannah Campus, for taking care of the bibliography. Darcy Lima passed away in July 2015 and I wanted to acknowledge her infinite encouragement to write this and the many other papers, book chapters, and books we wrote together. He will be forever remembered.

**Conflicts of Interest:** The author states that there are no conflicts of interest.

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
