# Peer review of "Preliminary Studies on Genetic Profiling of Coffee and Caffeine Consumption"

_beverages, doi:10.3390/beverages5030041_

Round 1
Reviewer 1 Report
- An interesting paper on genetic analysis depicting how the 16 known SNPs are expressed in both in terms of frequencies of mutant alleles and intra- and inter-chromosomal association following the consumption of coffee and caffeine. However, one would have expected some detailed explanation on how the possible linkage between this information and the various chronic disorders which coffee and caffeine consumption seem to prevent can be extrapolated.
- There is some repetition in information contained in the titles of some of the figures which happens to be identical to that in the text e.g. Title in Figure 2 and sentence in lines 117-119.
- Is it possible to establish from the available data if there is significant correlation between the level of caffeine in the blood samples and the number of mutants per volunteer?
- There is one correction in the text see line 173, check whether it is valid.

Author Response
1) an explanation on how the possible linkage between our data and the various chronic diseases which coffee /caffeine intake seems to prevent - It is our expectation for our next study with a broader population comparing equivalent groups of volunteers that does and does not consume coffee/caffeine;
2) The repetition in the text and the label of the figure was removed. Figure 2 was deleted, considering the replacement of another table (Table 3) which is more informative than the figure 2.
3)Correlation between caffeine plasma levels and and presence of variant alleles- we found two interesting correlations: the volunteer with the lowest level of caffeine in plasma and phenotype of fast metabolizer was the one with the smaller number of variant alleles across the 16 SNPs. On the contrary, the volunteer with the highest level of caffeine in plasma and phenotype of slow metabolizer was the one with highest number of variant alleles across the 16 SNPs (Discussion).
4) Line 173 was corrected.
Reviewer 2 Report
Dr Santo presents a descriptive study of caffeine/coffee-associated SNPs among 13 healthy volunteers. I have many concerns with the paper.
1. The objective of the study was to examine relationships between 16 SNPs previously linked to coffee/caffeine consumption. What was the purpose of the plasma caffeine measure? Why was this included at all?
2. Of the 16 SNPs examined, several of these are in well-known linkage disequilibrium (LD). The latter has been defined by the HapMap project and all data is available and has been used by geneticists for years. The objective of the study and entire results section describes this linkage in terms of ‘associations’. The objective of the study has therefore been captured within a previous genome-wide consortium effort and the results are not new.
3. The paper is missing a statistical analysis section.
4. Table 1 needs references. Why was ADORA2A SNP included? The last column of this Table is unclear and contrasts with genes listed in column 5.
5. Data is presented for 13 subjects but the abstract states 15 subjects. Presentation of subject specific data is not informative.
6. Results: figure 1 is actually a table. Last row: what does “OR” mean? The ‘mutant’ terminology is not appropriate for this study. The number of “mutants” (variants) is also not informative (see next comment)
7. The total number of ‘mutants’, minor alleles, across SNPs is meaningless as the direction of effect for each allele on the trait of interest (i.e. caffeine intake) differs.
8. Figure 2 is not clear and is also not informative given known and published LD among SNPs.
9. I believe Figure 3 and Figure 4 refer to the same Figure. This figure and all ‘results’ text suggests the author did the experiments and these are the ‘results’. This is actually work by others and appropriate only for the discussion.
Author Response
1) The caffeine plasma levels were part of the data from the pilot study. I thought it would be useful to include information of caffeine plasma levels and their genotype for caffeine metabolizing polymorphic enzyme CYP1A2. It actually allowed to conclude that the volunteer with highest level of caffeine and also a show caffeine metabolizer was the only one with the higher number of variant alleles within the 16 SNPs. As opposed to the volunteer with the lowest caffeine plasma levels, who was the one with the smaller number of variants within the same 16 SNPs.
2) The new information is the inter-relation between genotypes across the group of volunteers. We identified some interesting patterns, most of them presenting similar data, which we called positive association and also a few ones with opposite patterns; which we called negative association;
3) We think that because our sample is so small we preferred to just make a qualitative analysis of the results, in terms of similar and dissimilar patterns of expression of the genotypes;
4) ADORA2A SNP was included as part of the tentative to look for correlations related to the pharmacodynamics of caffeine as well;
The last column was deleted, since we already talk about the association in the text and show in the figures;
5) There were couple of samples that were mishandled during test preparation in the laboratory and we would not have enough DNA to run all the assays, so we excluded them ( this is now mentioned in the manuscript)
6) The figure 1 was now labeled as Table 2
7) I am interested not in the quantity (total number of mutants/variants) but if they are present or not in the same volunteers (trends and patterns)
8) Figure 2 was removed from the manuscript and instead a new table (Table 3) was added to facilitate to understand the similar and opposite associations suggested by our data;
9) As I mentioned in the previous note (8) figure 2 was removed and just the figure with the alignment of the genes/SNPs was kept to show how intricate is the cross-talk between chromosomes/genes.
Reviewer 3 Report
I wish the authors for their time in preparing the manuscript "Preliminary studies on genetic profiling of coffee and caffeine consumption. Introduction: There are missing references and reference inconsistencies throughout the introduction. I encourage the authors to revise and amend these where appropriate. Methods: Can you please clarify why in the first study (pilot), genotyping was only performed on 11 samples due to "satisfactory DNA quality" only obtained in these samples as stated in reference 7 but in the current study you included 13. Was ethical approval obtained? Please include a statement in the manuscript. Please re-phrase lines 83-85. This is confusing and somewhat backwards in the way HPLC analysis is usually written. Results Line 109 - provide appropriate figure number Line 110-111 - This would be more appropriate in the discussion section Lines 117-119 - This information is more appropriate to be described in the discussion along with figure 2. These are not results of the current study rather a summary of the functions of the SNPs found in the current study. References are required for figure 2 legend. Lines 131-138 - More suited to the discussion along with figure 3 Line 131 - you refer to figure 4 however there is no figure 4 How did you determine the associations of the genotype (line 149). Please explain. Again in line 165 forward - please justify the use of positive and negative associations and how these were determined. Line 208-209 - Please ensure correlations to phenotype and plasma caffeine concentrations are reported in the results section. Also, this statement is contradictory to the statement made in the introduction (lines 41-45) speaking about the previous publication utilising the same study population.
Author Response
Introduction -References inconsistencies were corrected
Methods; in the pilot study we had problems during the execution of the assay, at that time only for CYP1A2 and we could not reproduce the data, reason why we only run 11 instead of 13. In this new study we were able to fix the problem of DNA quality, mentioned in the pilot study and all the 13 viable samples (out of the initial total of 15) were utilized.
The previous pilot study, as well as the new study were all submitted to the South university Review Board and are approved. A statement was added to the manuscript.
Lines 83-85 : HPLC method of analysis of caffeine in the plasma was reedited.
Results line 109 - number of the figure corrected
Lines 110-111 Fixed
Lines 117-119 Removed
Figure 2 is now Table 2 and the information contained there is official from NCBI database.
Lines 131-138 - The inconsistencies ere fixed ( figure numbers )
Line 149 - This study due to a very small sample,we only looked at qualitative results. We found a trending in the expression of variants over wild-type and vice-versa. We called positive association when the pattern of the genotypes across the same group of volunteers was predominantly similar and we called negative (inverse or opposite) association when the pattern seen was inverted between the SNPs.(Line 165)
Lines 208-209 - The contradiction was fixed and the levels of caffeine are displayed on Table 2 (previous Figure 2) on the first column and the genotype for caffeine metabolism is the SNP # 9 in the same table.
Lines 41-45 - The new manuscript suffered a major revision and I believe with the new editing all the concerns and doubts are clarified.
Round 2
Reviewer 2 Report
See attachment

Author Response
1) This point was addressed on the second round of revision. The statement was changed and it is not showing as "conclusive' but just mentioned as an interesting finding, due to small number of subjects studied.
2) Addressed in the introduction section (lines 47-58).
3) I did find support for my data, comparing my results with a recent GWAS of Coffee Consumption.
Addressed in the abstract section (lines 17-20) and conclusion (lines 235-241)
4) Addressed in the previous round of revision
5) Addressed as the manuscript had a important revision in terms of correct terminology and conclusions.
6) Table 2 has the terminology changed for a more appropriate terms,as requested.
7) Table 3 was entirely revisited with new display of the data. It was added the variant-allele effect direction based on the frequencies found for the 3 possible genotypes.
8) Figure 2 was removed.
9) The figures were removed and the results and discussion section were changed accordingly.
Reviewer 3 Report
Thank you for making the requested changes.
Author Response
Dear Reviewer,
Thank you for taking time to read and critique my manuscript.
Roseane Santos